# Honey’s Yeast—New Source of Valuable Species for Industrial Applications

**DOI:** 10.3390/ijms24097889

**Published:** 2023-04-26

**Authors:** Patrycja Ziuzia, Zuzanna Janiec, Magdalena Wróbel-Kwiatkowska, Zbigniew Lazar, Magdalena Rakicka-Pustułka

**Affiliations:** 1Department of Biochemistry and Molecular Biology, Wroclaw University of Environmental and Life Sciences, 31 Norwida St., 50-375 Wroclaw, Poland; patrycja.ziuzia@upwr.edu.pl; 2Department of Biotechnology and Food Microbiology, Wroclaw University of Environmental and Life Sciences, 37 Chełmońskiego St., 51-630 Wroclaw, Poland

**Keywords:** kynurenic acid, erythritol, mannitol, citric acid, honey, yeast

## Abstract

Honey is a rich source of compounds with biological activity; moreover, it is a valuable source of various microorganisms. The aim of this study was to isolate and identify yeast from a sample of lime honey from Poland as well as to assess its ability to biosynthesize value-added chemicals such as kynurenic acid, erythritol, mannitol, and citric acid on common carbon sources. Fifteen yeast strains belonging to the species *Yarrowia lipolytica*, *Candida magnolia*, and *Starmerella magnoliae* were isolated. In shake-flask screening, the best value-added compound producers were chosen. In the last step, scaling up of the culture in the bioreactor was performed. A newly isolated strain of *Y. lipolytica* No. 12 produced 3.9 mg/L of kynurenic acid growing on fructose. Strain *Y. lipolytica* No. 9 synthesized 32.6 g/L of erythritol on technical glycerol with a low concentration of byproducts. Strain *Y. lipolytica* No. 5 produced 15.1 g/L of mannitol on technical glycerol, and strain No. 3 produced a very high amount of citric acid (76.6 g/L) on technical glycerol. In conclusion, to the best of our knowledge this is the first study to report the use of yeast isolates from honey to produce valuable chemicals. This study proves that natural products such as lime honey can be an excellent source of wild-type yeasts with valuable production properties.

## 1. Introduction

Honey is a natural sweet substance collected and transformed by bees (*Apis mellifera*). The starting material for honey production is nectar, the secretions of living parts of plants or excretions of plant-sucking insects residing on the living parts of plants (Council Directive 2001/110/EC, 2002). Regarding its high nutritional and health-related properties, honey has been widely used since antiquity to treat superficial wounds and intestinal diseases [1]. Furthermore, this natural sweetener has been proven to have antioxidant [2], anti-inflammatory [3,4], and anti-cancer activities [5]. The composition of natural honey distinguishes two basic main ingredients, namely, carbohydrates and water. Depending on the origin and climate as well as on the type of plants from which the secretions were collected, honey can differ significantly in its quantitative and qualitative content.

Honey is largely composed of glucose and fructose (about 75%), and the quantitative ratio of these two sugars is often an indicator of its origin. In most types of honey, the amount of fructose slightly exceeds the amount of glucose. Exceptions to this rule are grape honey (*Brassica napus*) and honey obtained from daffodils (*Taraxacum officinale*) [6]. The second most abundant component of honey is water, which is a factor determining the occurrence of microorganisms. The water activity in honey oscillates around the level of 0.5–0.65 [7]. Moreover, the protein content and qualitative composition of amino acids varies depending on the origin of the honey [8]. The most common amino acid is proline (50–85% of the total amount of amino acids), followed by glutamic acid, alanine, phenylalanine, tyrosine, leucine, and isoleucine [6,8]. Among the organic acids found in honey, the highest concentration (approximately 0.57%) has been measured for gluconic acid; it significantly affects the pH of the honey, which ranges from 3.2 to 4.5 [6]. Potassium (K) is the main mineral component found in honey, though significant amounts of sodium (Na), iron (Fe), copper (Cu), silicon (Si), manganese (Mn), magnesium (Mg), and calcium (Ca) have been reported in the literature as well [6]. Honey contains vitamin C and a complex of B vitamins (B2, B3, B5, B6, B8, B9), which are protected from oxidative degradation due to the low pH [6]. Honey has antibacterial and antifungal properties as well, which are mostly caused by the low water activity, presence of gluconic acid, and presence of fatty acids originating from royal jelly, the larval food of honeybees [9]. The aliphatic C8–C12 acids, which have been identified in honey samples, demonstrate antibacterial action against different microbes, and are the main components exhibiting antibacterial activity in honey [9]. Another compound with this activity in honey is phenolic acid, which is present in the highest concentrations in phacelia (356.72 µg/g) and multifloral honeys (318.9 µg/g) [10].

The physicochemical properties of honey affect microbial development. Due to the low water activity and low pH, mostly caused by gluconic acid, as well as the presence of antimicrobial components such as hydrogen peroxide, the growth of most microorganisms is inhibited [11,12]. Although the crucial factor in honey’s antimicrobial properties is low water activity (0.5–0.65), this is also the crucial factor in the ability of microbes to grow in such an environment. A number of microorganisms, especially spore-forming bacteria and sugar-tolerant yeasts, manage to survive the osmotic stress and low pH of honey. The yeast species most commonly isolated from this environment are *Ascophaera* spp., *Aureobasidium pullulans*, *Candida* spp., *Cryptococcus uzbekistanensis*, *Debaromyces* spp., *Dioszegia* spp., *Hansenula* spp., *Lipomyces* spp., *Metschnikowia reukaufii*, *Nematospora* spp., *Oosporidium* spp., *Pichia* spp., *Rhodotorula* spp., *Saccharomyces* spp., *Schizosaccharomyces* spp., *Shwanniomyces* spp., *Sporobolomyces* spp., *Torula* spp., *Trichosporon* spp., *Vishriacozyna* spp., and *Zygosaccharomyces* spp. [11,13,14,15,16,17,18].

In addition to being a good source of interesting microorganisms, honey is a rich source of biologically active compounds, including kynurenic acid. This very important acid, synthetized endogenously in the human body through the tryptophan catabolic pathway [19], plays a significant role in multiple biological processes and is crucial for human well-being [20]. There are many reports in the literature indicating that kynurenic acid can act as an anti-proliferative factor towards cells involved in the tumorigenesis process, e.g., the colon adenocarcinoma cells Caco-2, HT-29, and LS-180 [21], as well as having an apoptotic effect on gastric cancer AGS cells [22]. Nevertheless, it is the neuroprotective properties of kynurenic acid that make this compound of great interest for further investigation [23]. Kynurenic acid has been described as an antagonist of some of the most widespread excitatory neurotransmitters in the central nervous system (CNS), specifically, ionotropic glutamic acid receptors such as N-methyl-D-aspartate (NMDA), α-amino-3-hydroxy-5-methyl-4-isoxazolepropionic acid (AMPA), and kainic acid (KARs) receptors [24]. Considering the fact that increased activity of these above-mentioned receptors may be a case of neurodegenerative disease development such as Alzheimer′s and Huntington′s diseases, the balance of the internal kynurenic acid level is extremely important [24,25,26].

Unfortunately, the main obstacles in using kynurenic acid therapeutically are its low water solubility and the limited amount of natural sources containing this compound [27,28]. Until now, low concentrations of kynurenic acid have been detected in meat, vegetables, fruits, dairy products, herbs, and spices [29]. Surprisingly, the highest amount of kynurenic acid has been detected in honey and other bee-related products, among which chestnut honey has the highest concentration of this compound [29,30]. An alternative way to provide kynurenic acid in the diet with natural food products may be its biotechnological production using yeast, such as *Saccharomyces cerevisiae*, *S. pastorianus*, and *Yarrowia lipolytica* [19,31,32,33]. Although previous research has shown *Y. lipolytica* to be a good producer of kynurenic acid, its intracellular function in yeast remains unknown [19,34].

Due to the unconventional nature of honey as a microbial environment with reduced water activity, the potential of yeast isolated from such environments has been analyzed towards biosynthesis of other value-added metabolites such as erythritol, mannitol, and citric acid [35,36]. Nowadays, most polyols produced by yeast are considered high value-added products, with an excellent example being erythritol, a four-carbon compound with no optical activity [37]. In nature, erythritol is present in small amounts in mushrooms and fruits, i.e., pears, grapes, and watermelons [37]. It tastes similar to sucrose, however, it has a glycemic index of zero, which makes it ideal for diabetics [38]. Currently, erythritol is produced on an industrial scale via biotechnological methods using *Moniliella pollinis*, *Trichosporonoides megachiliensis*, and *Y. lipolytica* [39]. In addition, the above-mentioned osmophilic yeasts are known to secrete mannitol, a six-carbon polyol. It can be found in many plants and algae; however, it is produced as a main polyol in fungal hyphae as well [40]. Due to its properties, mannitol is widely used in the chemical and pharmaceutical industries, and has become a component in functional food manufacturing [41,42]. Industrially, mannitol is produced using a multi-stage chemical synthesis process [42]; alternatively, biotechnological methods using microorganisms, mainly *Candida* spp., could be successfully used [42].

Another excellent example of biotechnological production of industrially valuable compounds is citric acid, a primary metabolite in the Krebs cycle. Citric acid is efficiently produced by the filamentous fungi *Aspergillus niger* as well as by certain yeasts, including *Candida* spp., *Hansenula* spp., *Pichia* spp., *Debaryomyces* spp., *Rhodotorula* spp., *Nocardia* spp., *Saccharomyces* spp., *Zygosaccharomyces* spp., and *Y. lipolytica* [43]. Utilization of yeast for citric acid production brings many advantages. At first, yeast have the ability to utilize various carbon sources, including waste products, which are often contaminated with metal ions. Yeasts are less sensitive to unfavorable parameters of technological processes, such as low oxygenation. Unfortunately, drawbacks exist as well, including the formation of undesirable iso-citric acid [43].

The aim of the present study was to isolate and identify new strains from lime honey (Legnica, Poland) as efficient producers of kynurenic acid, as well as other value-added metabolites such as erythritol, mannitol, and citric acid. Glycerol and fructose were used as carbon sources. In order to meet the needs to reduce environmental pollution and promote sustainable economics, technical glycerol was used as a carbon and energy source.

## 2. Results and Discussion

### 2.1. Yeast Isolation and Identification

The scheme of the conducted experiments is shown in Figure 1. New and potentially interesting yeast strains were isolated from lime honey samples using a medium containing chloramphenicol, which inhibits the growth of bacteria. Snowdon and Cliver [18] have reported that although yeast content in many honey samples is below 100 CFU per gram, this amount can nonetheless grow to reach very high numbers. In the present study, the total number of isolates reached 3.8 × 10^5^ CFU/g of honey. Fifteen yeast strains were selected for further analysis. The phenotypic characteristics of the selected strains are summarized in Table 1. All strains were of light beige color, and were grown as wrinkled or smooth colonies (Table 1).

Subsequently, three different methods of yeast identification were applied. All the isolated strains were identified using an API^®^/ID32 commercial identification kit, which was used according to the manufacturer’s protocol. Furthermore, MALDI TOF-MS analysis was performed to allow for the identification of microorganisms based on their unique ribosomal protein profiles [44]. The results obtained after sequencing the rDNA with both ITS1-NL4 primers were used to assign the analyzed strains to the appropriate species, as listed in Table 2.

Using a rapid kit for API^®^/ID32 yeast identification, isolates Nos. 1–7, 9, and 11–14 were identified as *Candida lipolytica* with a high degree of certainty (Table 2). Isolates Nos. 10 and 15 turned out to be *C. magnoliae*. Isolate No. 8 was recognized as *C. magnoliae* or *C. globosa* (Table 2). On the other hand, strains Nos. 1–7, 9, and 11–14 were identified as species of *Y. lipolytica* using the MALDI TOF-MS method (Table 2). Seven strains from this group were identified with a high degree of pattern matching (Table 2). Isolates Nos. 8, 10, and 15 were *C. magnoliae*. We further confirmed our findings by genetic identification. The majority of the analyzed 28S rDNA sequences showed identities of 98% similarity or higher with sequences from the GenBank database (http://www.ncbi.nlm.nih.gov/blast, accessed on 24 March 2023). These results confirmed that isolates Nos. 1–7, 9, and 11–14 were *Y. lipolytica* and that isolates Nos. 8 and 10 were *C. magnoliae* (Table 2). Using the sequencing method, isolate No. 15 was identified as a *Starmerella magnoliae* (Table 2), formerly known as *C. magnoliae*. The species *Y. lipolytica* and *S. magnoliae* could not be identified by the API^®^/ID32 test because they were not described in the available database. Similarly, MALDI TOF-MS does not have a description of *S. magnoliae* in the library. In this study, the identification methods provided comparable results and demonstrated the good reliability of the genetic identification and MALDI TOF-MS methods as routine techniques for the identification of honey yeast isolates.

In summary, the yeasts isolated from lime honey from Poland belonged to *Y. lipolytica* and *C. magnoliae/S. magnoliae*. *Y. lipolytica* yeasts are usually isolated from hydrophobic environments and contain oils [45]. For such a large number of *Y. lipolytica* yeasts to be isolated from honey is surprising and unusual for this environment. This type of yeast usually has few sugar transporters and a natural ability to utilize a small range of carbohydrates [46]. When compared to the results described in the literature, the yeasts most often isolated from honey belong to the following genera: *Saccharomyces* spp., *Debaromyces* spp., *Hansenula* spp., *Rhodotorula* spp., *Lipomyces* spp., and *Candia* spp. [18]. The species of yeasts isolated from honey strictly depend on the type of honey and the region in which it is produced [47]; for example, the genera *Starmerella* and *Zygosaccharomyces* have been predominantly found in honey produced by stingless bees raised in the highlands of southern Brazil [47].

### 2.2. Growth of Yeasts Isolated from Honey on Different Carbon Sources

The capacity of yeast isolates to grow on different carbon sources was investigated using a Bioscreen C microplate reader. The isolates were grown in YNB medium supplemented by 2% of the corresponding carbon source (glucose, fructose, inulin, sucrose, and technical glycerol). The growth curves are presented in Figure 2. The main differences were observed among the isolates growing on glucose (Figure 2a). Strain No. 8 achieved the highest growth on glucose, while for strains Nos. 10 and 15 the growth was 7% lower compared to strain No. 8 on the same substrate. The remaining strains obtained lower growth, reaching as much as 28–40% of the maximum growth reached by strain No. 8 growing on glucose (Figure 2a). The analyzed *Y. lipolytica* strains isolated in this study showed weaker growth on glucose than the *Y. lipolytica* strain A101 described by Dobrowolski et al., 2019 [48]. Considering the second most abundant sugar, namely, fructose, the strains with the highest growth were Nos. 8, 10, and 15 (Figure 2b). On the contrary, strains Nos. 4, 7, and 11 showed lower growth by 41–48% compared to strain No. 8 growing on fructose (Figure 2b). The observed differences in growth between glucose and fructose may be connected to high oxygen demand for *Y. lipolytica*, considering that oxygen availability during microculture is limited. The *Candida/Starmerella magnoliae* isolates Nos. 8, 10, and 15 grew well on inulin and sucrose (Figure 2c,d). The other isolates did not grow on inulin and sucrose (Figure 2c,d), which confirms the general knowledge about *Y. lipolytica* strains, specifically, that they are not able to use sucrose or insulin as a sole carbon source [49]. The most significant differences were observed for the analyzed set of strains growing on technical glycerol (Figure 2e). Strain No. 1 did not grow at all on this medium. Isolate No. 10 showed the best growth and reached the highest biomass (Figure 2e). For strains Nos. 2–5, 7–9, 11–12, and 15, the growth on technical glycerol was lower by 15–20% compared to strain No. 10. Strains Nos. 6 and 13 showed longer lag phases compared to the other isolates and reached the lowest growth at the end of the culture (22–37% lower than for strain No. 10). The growth of *Y. lipolytica* strains Nos. 2–5, 7–9, and 11–12 on technical glycerol was similar to the growth of other *Y. lipolytica* strains described by Mirończuk et al., 2017 [50].

### 2.3. Value-Added Chemical Synthesis Using Yeast Isolated from Honey in Shake-Flask Screening

After characterizing the growth of the isolates on different media with common carbon sources, they were investigated for their abilities to produce value-added chemicals using the shake-flask culture method. Based on the obtained results, the best producers of kynurenic acid, erythritol, mannitol, and citric acid were selected.

The available published data show that high concentrations of kynurenic acid can be found in different types of honey [51]. For this reason, we investigated the capacity for kynurenic acid production by the yeast strains isolated from lime honey. The strains were grown in a shake-flask in medium with fructose as a carbon source and CaCO_3_ for pH control. Fructose was chosen as the carbon source based on our previous studies of kynurenic acid production using a *Y. lipolytica* S12 strain [31]. All tested strains produced kynurenic acid, with concentrations ranging from 0.3 mg/L for isolate No. 15 to 13.6 mg/L for isolate No. 12 (Figure 3a). High kynurenic acid concentration was inversely correlated with low biomass concentration (Figure 3b). This phenomenon confirms previously observed results of kynurenic acid biosynthesis by different *Y. lipolytica* strains [34]. The 13.6 mg/L of kynurenic acid produced by strain No. 12 is higher than the amount reported for *S. cerevisiae* and *S. pastorianus* by Yılmaz and Gökmen [32]; however, it is lower than that observed for the *Y. lipolytica* S12 strain in our previous research [31].

Subsequently, the production of polyols and citric acid by the new isolates were investigated using the shake-flask experiment. The composition of media used in this experiment were selected based on our previous studies [52,53,54]. ERYMed, MANMed, and CAMed media were optimized for *Y. lipolytica* to secrete erythritol, mannitol, and citric acid, respectively. Interestingly, the analyzed strains did not produce the expected compounds in the dedicated media (Figure 4). For example, in ERYMed medium optimized for erythritol production, the isolates produced high concentrations of citric acid and had weak production of erythritol (Figure 4a). In MANMed medium, the tested yeasts produced moderate amounts of mannitol and higher production of citric acid (Figure 4b). Finally, in CAMed medium, which is dedicated to citric acid production, the concentration of erythritol was favored, with only moderate citric acid production (Figure 4c). The high production of citric acid, noted almost for all investigated yeast isolates in the ERYMed and MANMed media, may be related to the natural abilities of these strains, too low a C:N ratio in the media, or too long of a culture time (168 h) [55]. The exhaustion of glycerol in the media resulted in the yeasts consuming other available carbon sources, namely, erythritol and mannitol, simultaneously with increasing production of citric acid. The highest concentration of erythritol in the ERYMed medium was noted for isolate No. 9 (7.5 g/L) (Figure 4a). All tested yeasts grew well in the ERYMed medium, which is supplemented with NaCl to induce high osmotic pressure to promote the production of erythritol [55]. The highest mannitol production in the MANMed medium was observed for isolate No. 5 (13.5 g/L) (Figure 4b). Strain No. 10 was the only one that did not produce mannitol in these conditions (Figure 4b). Erythritol was produced by almost all strains in the MANMed medium, despite the lack of NaCl supplementation in this medium (Figure 4b). When CAMed was applied, the highest concentration of citric acid was secreted by isolate No. 3 (5.7 g/L) (Figure 4c). Citric acid is produced by *Y. lipolytica* under conditions of limited nitrogen availability [36]. Surprisingly, in the CAMMed medium used in this study, the analyzed strains did not produce high amounts of citric acid; however, they produced large amounts of polyols instead (Figure 4c). The CAMMed medium composition consists of a high quantity of yeast extract (3 g/L), which might favor the biosynthesis and secretion of polyols by the tested yeasts. In subsequent studies, it would be worthwhile to assess the effects of yeast extract concentration on citric acid/polyols synthesis by the chosen strains as well as optimize the medium composition and culture conditions for particular strains. These conditions caused increased biomass production, which in turn increased the concentration of polyols. No residual glycerol was observed in any cultures. None of the shake-flask cultures presented in Figure 4 contained residual glycerol. Both the microculture and shake-flask experiments showed that the tested isolates were highly sensitive to oxygenation of the medium. While the growth of yeast on glycerol was weak in the microcultures (Figure 2e) due to limited oxygenation, the same strains grew very well on glycerol in the shake-flask cultures (Figure 3). The shake-flask screening for value-added chemical biosynthesis allowed us to choose the best potential producers of these compounds for subsequent scaled up bioreactor processes.

### 2.4. Bioreactor Studies

The best candidate strains chosen in the previous steps were analyzed for their production of valuable capacity during bioreactor cultures. Kynurenic acid, erythritol, mannitol, and citric acid secretion were investigated. The cultures were performed in 5 L bioreactors with a working volume of 2 L.

Isolate No. 12 was chosen as the best kynurenic acid producer for large scale production. Fructose was applied as the carbon source and the culture lasted 168 h (Figure 5a). Strain No. 12 produced 3.9 mg/L of kynurenic acid, which is lower than that obtained in the shake-flask experiment (13.6 mg/L) (Figure 3a and Figure 5a). The productivity for kynurenic acid was around 0.02 mg/L × h and the yield reached 0.1 mg/g (Table 3). The biomass concentration at the end of the process reached 17.4 g/L; however, it started to decrease after 120 h (Figure 5b). These results show that scaling up and further optimization of the culture parameters is necessary in order to increase the secretion of kynurenic acid. One possibility might include tryptophan supplementation. The available literature provides evidence that supplementation of the medium with tryptophan (200–400 mg/L) can increase the concentration of kynurenic acid for *S. cerevisiae* and *Y. lipolytica* [31,32,56].

For erythritol production in the bioreactor, strain No. 9 was chosen. In a 95-h culture this yeast produced 32.6 g/L of erythritol, which was higher during the shake-flask experiment. The results obtained for other yeast species belonging to the *Yarrowia* clade (*C. oslonensis*, *Y. divulgata*, and *C. hollandica*) growing in the same medium reached 44.6 g/L, 35.4 g/L, and 33.4 g/L of erythritol, respectively [54]. Similar yields of erythritol have been achieved for other species, for example, *Moniliella pollinis* growing on sugarcane juice in a fed-batch culture, where erythritol yield reached 0.38 g/g and its productivity was 0.61 g/L × h [57]. In turn, *Trichosporonoides oedocephalis* was able to produce erythritol with a 0.35 g/g yield growing on glucose [58]. The advantage of erythritol production using the newly isolated *Y. lipolytica* No. 9 strain was the low concentration of arabitol and mannitol in the culture. The production of mannitol and arabitol reached 4.8 g/L and 1.3 g/L, respectively (Figure 5b, Table 3). The productivity and yield of erythritol were 0.34 g/L × h and 0.33 g/g, respectively (Table 3). The analyzed strain was not sensitive to NaCl supplementation, which improved the purity of erythritol biosynthesis in the particular medium composition. The biomass concentration in this process reached 20.75 g/L (Figure 5b). Furthermore, the secretion of citric acid was low and reached only 1.1 g/L (Table 3). Comparing the results obtained for *Y. lipolytica* strain No. 9 isolated from honey to the results described in the literature for another *Yarrowia* clade species, the new isolate turned out to be good producer of erythritol. However, further process development with this strain is required.

Mannitol production was performed in the bioreactor using *Y. lipolytica* strain No. 5. In this condition, the production of mannitol was low, reaching only 15.1 g/L, with productivity and yield of 0.16 g/L × h and 0.15 g/g, respectively (Figure 5c, Table 3). The biomass concentration reached 19.75 g/L; however, what was very surprising in this process was the very high quantity of citric acid secreted (30.2 g/L) (Figure 5c). Erythritol secretion in this condition was not observed. Until now, the best producer of mannitol described in the literature has been *C. magnoliae* growing in a fed-batch culture, reaching a mannitol concentration in the medium of 209 g/L [59]. To date, there have been no efficient mannitol producers described among *Y. lipolytica* strains which can secrete mannitol with high purity. Only simultaneous production of erythritol and mannitol by various strains of *Y. lipolytica* has been reported [40,60].

For our last experiment involving citric acid biosynthesis, *Y. lipolytica* No. 3 was chosen as the best candidate strain. The high C:N ratio of the production medium favored mostly citric acid biosynthesis; during a 120-h culture, this strain produced 75.7 g/L of citric acid (Figure 5d). The productivity and yield reached 0.63 g/L × h and 0.76 g/g, respectively (Table 3). During citric acid biosynthesis, polyol formation was very low, and reached only 7.15 g/L of mannitol, 4.05 g/L of erythritol, and 2.05 g/L of arabitol (Figure 5d). As expected, the biomass concentration reached only 12.5 g/L (Figure 5d). The results described in this study are mostly higher than those obtained for other *Y. lipolytica* strains growing on glucose [61]. Previously tested *Y. lipolytica* strains produced citric acid with yields varying from 0.38 g/g to 0.85 g/g on glucose [61]. The lowest reported citric acid yield for *Y. lipolytica* growing on glucose was noted for the NBRC 1658 strain (0.38 g/g) during batch culture and the H222 strain (0.39 g/g) growing in a fed-batch culture [61]. On the contrary, the highest citric acid yield was reached for *Y. lipolytica* strains growing on glucose; A-101 (repeated batch mode), W29 (batch culture), and A-101–1.14 (batch culture) reached 0.84, 0.85, and 0.93 g/g, respectively [61]. The citric acid yield obtained in this study for *Y. lipolytica* strain No. 3 during batch culture on technical glycerol (0.76 g/g) was higher than the highest yield (0.62 g/g) described to date in the literature for *Y. lipolytica* 1.31 growing on raw glycerol in a batch culture [61].

## 3. Materials and Methods

### 3.1. Honey Sample

Lime honey samples were aseptically obtained from the “Pod Starym Dębem” apiary in Legnica, Poland (51°12′36″ N; 16°09′42″ E), a region of Lower Silesia district in Poland. It is characterized by its light color, accentuated lime aroma, and intensely persistent odor.

### 3.2. Microorganism Isolation

Honey samples were weighed and dissolved in sterile H_2_O at a 1:9 ratio. The solution was diluted (from 10^−1^ to 10^−4^) and spotted on isolation medium: glucose 20 g/L (Merck, Darmstadt, Germany), yeast extract 3 g/L (Merck, Germany), malt extract 3 g/L (Merck, Germany), bactopeptone 5 g/L (Merck, Germany), agar 20 g/L (Merck, Germany), chloramphenicol 0.1 g/L (Merck, Germany). Plates were incubated for 48 h at 28 °C. Subsequently, based on the colony phenotype (smooth or wrinkled), 15 strains were selected. The yeast strains were stored in glycerol solution 25% (*v*/*v*^−1^) at −80 °C. Before use, liquid or plated YPD (glucose 20 g/L, yeast extract 3 g/L, malt extract 3 g/L, bactopeptone 5 g/L, agar 20 g/L only for plates) were inoculated and incubated for 24 h at 28 °C to refresh the yeast biomass.

### 3.3. Yeast Identification

Yeasts included in this study were identified by using three unbiased methods: biochemical (API^®^/ID32, Biomerieux, Marcy l’Etoile, France), MALDI TOF/MS (Jagiellonian Center of Innovation, Kraków, Poland), and rDNA sequence-based identification.

### 3.4. API^®^/ID32

The 24-h inoculum of the yeasts was added to API^®^ Suspension Medium and the density was standardized to a level of 2.0 according to the McFarland scale. In the next step, cell suspension was mixed with API^®^ C Medium, homogenized, and plated on API^®^/ID32 test string wells. Yeasts were cultured for 24 and 48 h at 28 °C and the results were obtained based on visual examination according to the presence or absence of turbidity in each well. Collected data were then transformed into numerical code and determined using APIWebTM (Biomerieux, Marcy l’Etoile, France) software.

### 3.5. MALDI TOF/MS

Yeasts were cultivated in YPD solid medium on Petri dishes and sent for analysis after 24 h. MALDI-TOF/MS was outsourced to the Jagiellonian Center of Innovation, Kraków, Poland.

### 3.6. rDNA Sequence-Based Identification

Genetically, the strains were identified using ITS1 (5′-tccgtaggtgaacctgcgg-3′) and NL4 (5′-ggtccgtgtttcaagacgg-3′) primers, which cover the whole rRNA coding region including the D1/D2 domain of the large subunit of rDNA as well as the ITS (internal transcribed spacer) domain (and the intervening 5.8S rRNA gene) [62]. DNA was extracted from cells growing on YPD medium for 24 h using the protocols of Hoffman and Winston [63]. Phire Plant Direct PCR Master Mix was used for the PCR reaction (Thermo Scientific, Waltham, MA, USA). The samples were run in 1% agarose electrophoresis and the obtained bands were extracted from the gel using a Gel-Out kit (A&A Biotechnology, Gdańsk, Poland). The amplicons were sequenced in both directions using the primers mentioned above (Genomed S.A., Warsaw, Poland). Results obtained from sequencing were analyzed using the Basic Local Alignment Search Tool (BLASTN) available in the public domain.

### 3.7. Microcultures

Growth of the analyzed yeast isolates on different carbon sources was analyzed using a Bioscreen C spectrophotometer (Oy Growth Curves Ab, Ltd., Turku, Finland). The 24-h yeast biomass was collected and washed three times in sterile distilled water, and the optical density (OD_600_) was standardized to a starting OD_600_ equal to 0.15. The cultures were carried out in growth medium consisting of Yeast Nitrogen Base 68 g/L (Sigma-Aldrich, Burlington, MA, USA) with 2% *w*/*v* of a suitable carbon source (glucose (Merck, Germany), fructose (Merck, Germany), inulin (Młyn Oliwski, Gdańsk, Poland), sucrose (Merck, Germany), or technical glycerol (Poch, Gliwice, Poland)) in a volume of 150 µL in 100-well culture microplates. The experiment was carried out at 28 °C with high constant shaking (1100 rpm). The growth curves were monitored by measuring the optical density (λ = 600 nm) every 30 min for 48 h. The assay was performed in triple biological replicates for each strain and the appropriate carbon source.

### 3.8. Media Compositions for Shake-Flask and Bioreactor Experiments

In this study, shake-flask and bioreactor experiments were performed to test the production of kynurenic acid, erythritol, mannitol, and citric acid.

The inoculation medium for microcultures in the shake-flask experiment and bioreactor cultures contained glucose 20 g/L, yeast extract 3 g/L, malt extract 3 g/L, and bactopeptone 5 g/L.

For kynurenic acid production, the medium consisted of fructose 40 g/L (Merck, Germany), (NH_4_)_2_SO_4_ 9 g/L, MgSO_4_ × 7 H_2_O 0.3 g/L, KH_2_PO_4_ 0.25 g/L, yeast extract 1 g/L (Merck, Germany), and CaCO_3_ 0.075 g/L (added 24 h after starting shake-flask experiment only) in 1 dm^3^ tap water (pH for the shake-flask experiment was 6.8, and for the bioreactor was 4.5 during the first 48 h and then 5.4) [31]. The media for erythritol production (ERYMed) for both shake-flask and bioreactor cultures were composed of technical glycerol (Poch, Poland) 100 g/L, (NH_4_)_2_SO_4_ 2.3 g/L, MgSO_4_ × 7 H_2_O 1 g/L, KH_2_PO_4_ 0.22 g/L, yeast extract 1 g/L, NaCl 26.6 g/L, and CaCO_3_ 0.075 g/L in 1 L tap water (only for the shake-flask experiment) (pH = 3.0) [35]. For mannitol production, the following medium composition (MANMed) was used: technical glycerol (Poch, Poland) 100 g/L, (NH_4_)_2_SO_4_ 2.7 g/L, MgSO_4_ × 7 H_2_O 1 g/L, KH_2_PO_4_ 0.20 g/L, yeast extract 1.6 g/L, CaCO_3_ 0.075 g/L (only for the shake-flask experiment) in 1 L tap water (pH = 3.0) [54]. The citric acid medium (CAMed) consisted of: technical glycerol (Poch, Poland) 100 g/L, NH_4_Cl 0.5 g/L, MgSO_4_ × 7 H_2_O 0.1 g/L, KH_2_PO_4_ 0.22 g/L, and yeast extract 3 g/L in 1 L phosphate buffer (pH 5.7 for the shake-flask experiment and pH 5.5 for bioreactor production) [36].

### 3.9. Culture Conditions for Inoculation, Shake-Flask and Bioreactor Experiments

For API^®^/ID32 testing, yeasts were cultured in 5 mL inoculation medium in 15 mL tubes on a rotary shaker (Elpan 358S, Poznań, Poland) at 28 °C, 180 rpm/3 Hz for 24 h. For MALDI TOF/MS analysis, yeasts were cultured on solid medium in Petri dishes at 28 °C in a laboratory incubator (Memmert INE 400, Chojnów, Poland) for 24 h. For the shake-flask experiments and bioreactor cultures, each yeast isolate was cultured in 50 mL inoculation medium in a 0.25 mL flask on a rotary shaker (Elpan 358S, Poland) at 28 °C, 180 rpm/3 Hz for 48 h. For the bioreactor cultures, the inoculum represented 10% of working volume.

The ability of the isolated yeasts strains to produce kynurenic acid, erythritol, mannitol, and citric acid was tested in shake-flask experiments using 25 mL of each production media in 0.25 mL flasks, with an initial OD_600_ 0.25 on the rotary shaker (CERTOMAT IS, Sartorius Stedim Biotech) at 28 °C, 200 rpm/3.3 Hz for 7 days. Each experiment was conducted in three biological replicates.

In the bioreactor experiments, the media were inoculated using the 48-h inoculum (washed three times with sterile distilled water) to the initial optical density of OD_600_ 0.5 and cultured at 28 °C, with aeration and rotary parameters equaling 0.8 vvm and 800 rpm/13.33 Hz, respectively. All cultures were performed in 5 L stirred-tank reactors (BIOSTAT B-PLUS, Sartorius, Germany) with a working volume of 2 L. During erythritol, mannitol, and citric acid production the pH was regulated with 20% (*w*/*v*) NaOH, while for kynurenic acid production 20% (*w*/*v*) NaOH and 15% (*v*/*v*) HCl were used. Each culture was performed in three biological replicates and continued until complete carbon source consumption.

### 3.10. Analytical Methods

Samples from shake-flask assays were taken aseptically at the end of each procedure. Media from each flask were diluted to a final volume of 25 mL, vortexed, and 10 mL of the solution was taken for further analysis. Samples (10 mL) from bioreactor experiments were taken continuously after 24 h. For both the shake-flask and bioreactor cultures, samples were centrifuged (Centrifuge MPW-56, Warsaw, Poland) for 5 min at 5000 rpm/83.3 Hz. Previously centrifuged biomass was separated from the supernatant using membrane filters ϕ = 0.45 µm (Merck Milipore, Burlington, MA, USA), dried to constant weight, and analyzed in a weight dryer (RADWAG MAC 110/NH, Poland) at 105 °C. Biomass was expressed in grams of cell dry weight per 1 L.

Kynurenic acid determination was conducted using a previously described protocol [31]. Supernatants were analysed using high-performance liquid chromatography (HPLC) (Dionex, UltiMate 8000 Series, Osaka, Japan) with a HYPERSIL Gold C18 column (Thermo Scientific^TM^, Waltham, MA, USA) connected to a DAD detector (λ = 210 nm) (Dionex-Thermo Fisher Scientific, Cheshire, UK). The column was eluted with 15 mM phosphate buffer (pH = 6.4) and 2.7% acetonitrile with a rate of 1.2 mL/min at 30 °C. The injection volume was 10 µL and the run time lasted 15 min. Data were calculated according to the kynurenic acid standard (Merck, Poznań, Poland) using Chromeleon software (Chromeleon^TM^ Chromatography Data System (CDS)).

Previously obtained and prepared supernatants were diluted at a 1:9 ratio with sterile H_2_O. Next, samples were analysed using HPLC with a HyperRez Carbohydrate H+ column coupled to a UV (λ = 210 nm) and refractive index (RI) detector (Shodex, Ogimachi, Japan). The column was eluted with 25 mM trifluoroacetic acetic acid with a rate of 0.6 mL/min at 65 °C. Data were analyzed according to the appropriate standard (glycerol, erythritol, mannitol, or citric acid) using Chromeleon software (Chromeleon^TM^ Chromatography Data System CDS)).

### 3.11. Fermentation Parameter Calculations

Productivity of biomass (Q_X_) was expressed in g/L·h and calculated using Equation (1)
(1)$$Q_{X}=\frac{\Delta X}{\Delta t}$$
where:$$Q_{X}-\mathrm{biomass}\;\mathrm{productivity},\;\left(\frac{g}{L}h\right)$$
$$\Delta X-\mathrm{biomass}\;\mathrm{increase}\;\mathrm{in}\;\mathrm{time},\;\left(\frac{g}{L}\right)$$
Δt−time of yeast cultivation,( h)

Productivity of metabolites Q_MET_ (kynurenic acid, erythritol, mannitol, and citric acid) was calculated using Equation (2) and expressed in g/L·h
(2)$$Q_{\mathrm{MET}}=\frac{\Delta P}{\Delta t}$$
where:$$Q_{\mathrm{MET}}-\mathrm{productivity}\;\mathrm{of}\;\mathrm{kynurenic}\;\mathrm{acid}\;\left(\frac{\mathrm{mg}}{L}h\right),\;\mathrm{erythritol},\;\mathrm{mannitol}\;\mathrm{and}\;\mathrm{citric}\;\mathrm{acid},\left(\frac{g}{L}h\right)$$
$$\Delta P-\mathrm{increase}\;\mathrm{of}\;\mathrm{kynurenic}\;\mathrm{acid}\;\left(\frac{\mathrm{mg}}{L}\right),\;\mathrm{erythritol},\;\mathrm{mannitol}\;\mathrm{and}\;\mathrm{citric}\;\mathrm{acid},\;\left(\frac{g}{L}h\right)$$
Δt−time of yeast cultivation, (h)

The yield of metabolites Y_MET_ (kynurenic acid, erythritol, mannitol, and citric acid) was calculated using Equation (3) and expressed in g/g
(3)$$Y_{\mathrm{MET}}=\frac{\Delta P}{\Delta S}$$
where:$$Y_{\mathrm{MET}}-\mathrm{yield}\;\mathrm{of}\;\mathrm{kynurenic}\;\mathrm{acid}\;\left(\frac{\mathrm{mg}}{g}\right),\;\mathrm{erythritol},\;\mathrm{mannitol}\;\mathrm{and}\;\mathrm{citric}\;\mathrm{acid},\;\left(\frac{g}{g}\right)$$
$$\Delta P-\mathrm{increase}\;\mathrm{of}\;\mathrm{kynurenic}\;\mathrm{acid}\left(\frac{\mathrm{mg}}{L}\right),\;\mathrm{erythritol},\;\mathrm{mannitol}\;\mathrm{and}\;\mathrm{citric}\;\mathrm{acid},\left(\frac{g}{L}\right)$$
$$\Delta S-\mathrm{substrate}\;\mathrm{loss},\;\left(\frac{g}{L}\right)$$

## 4. Conclusions

In the presented study, the potential of new yeasts isolated from lime honey from Poland for value-added chemical synthesis has been established. The wild *Y. lipolytica, C. magnoliae*/*S. magnoliae* strains were able to produce kynurenic acid, erythritol, mannitol, and citric acid in different concentrations growing on fructose or technical glycerol. Shake-flask screening allowed us to choose the best potential value-added chemical producing strains. The performed experiments on scaled-up bioreactor cultures provided particularly promising results for *Y. lipolytica* strains No. 9 and No. 3, which produced erythritol and citric acid in bioreactor cultures at relatively high concentrations comparable with the best data reported in the literature. Further analysis and process optimization are necessary to increase the yield and product titer of selected value-added chemicals using wild-type yeasts isolated from honey. One of the most interesting metabolites, which requires further intensive investigation, is kynurenic acid, which may attract considerable attention from the pharmaceutical industry. This study proves that natural products such as lime honey represent an excellent source of wild yeast strains with valuable production properties.

## Figures and Tables

**Figure 1 ijms-24-07889-f001:**
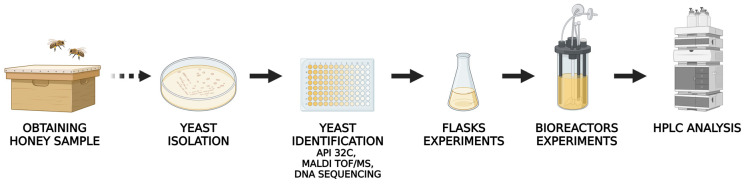
The scheme of the experiments performed in this work.

**Figure 2 ijms-24-07889-f002:**
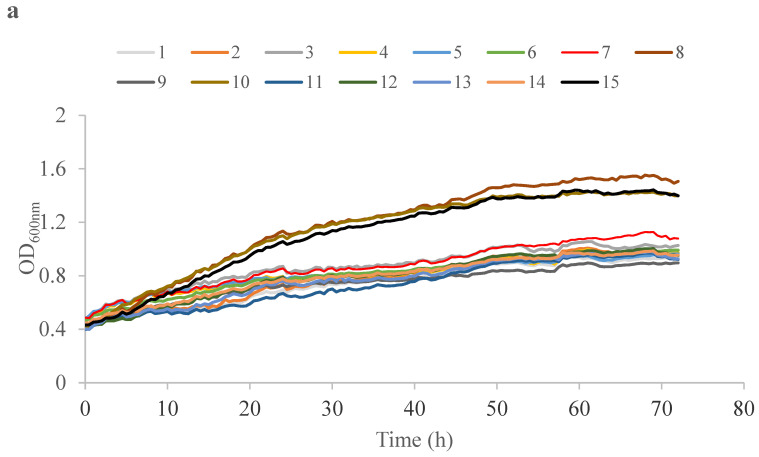
Growth curves of yeast strains isolated from honey (Nos. 1–15) growing on glucose (**a**), fructose (**b**), inulin (**c**), sucrose (**d**), and technical glycerol (**e**).

**Figure 3 ijms-24-07889-f003:**
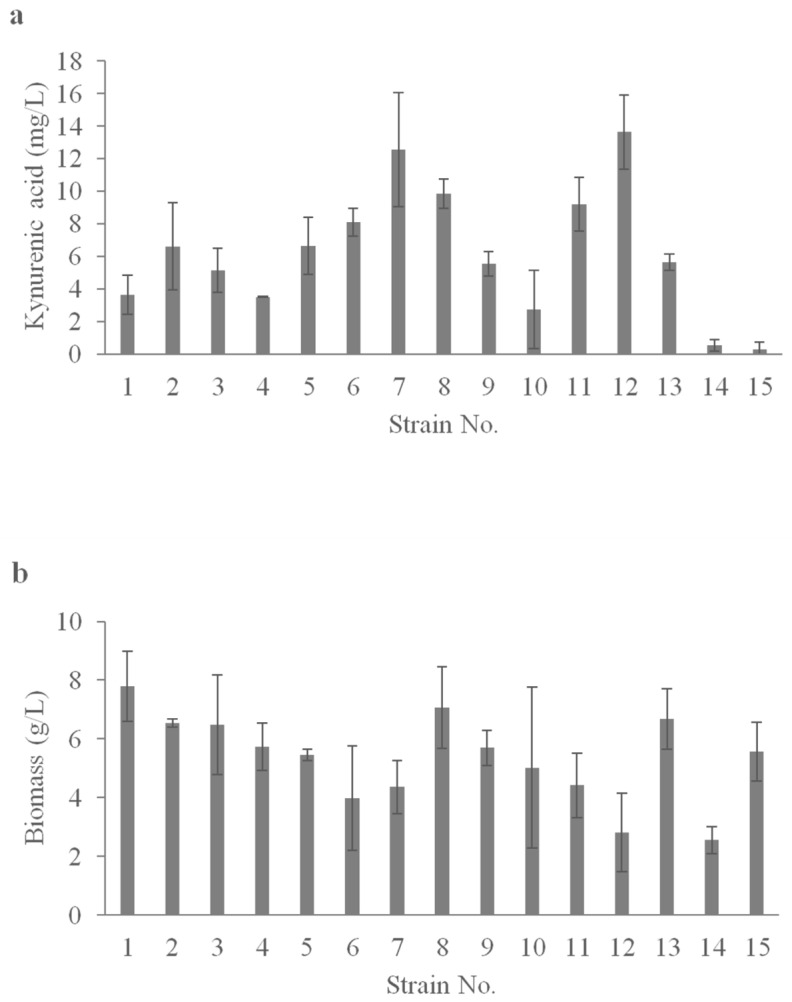
Kynurenic acid (**a**) and biomass (**b**) production in shake-flask experiment with yeast strains (No. 1–15) isolated from honey growing on fructose (40 g/L). The medium composition (g/L) was as follows: (NH_4_)_2_SO_4_–9, KH_2_PO_4_–0.25, MgSO_4_ × 7H_2_O–0.3, yeast extract–1.0, CaCO_3_–3.0 (added after 24 h of culture), tap water to 1 L, pH 6.8, 168 h, 28 °C, 200 rpm.

**Figure 4 ijms-24-07889-f004:**
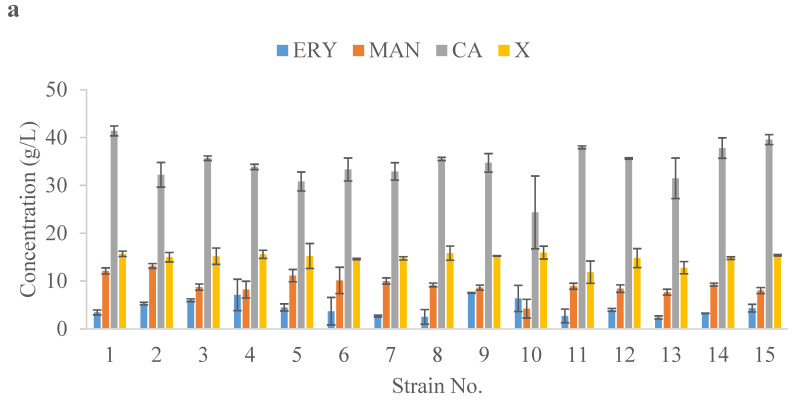
Erythritol (ERY), mannitol (MAN), citric acid (CA), and biomass (X) production in shake-flask experiment for strains isolated from honey (No. 1–15) growing in three different media: ERYMed (**a**), MANMed (**b**), and CAMed (**c**). The culture lasted 7 days at 180 rpm and 28 °C.

**Figure 5 ijms-24-07889-f005:**
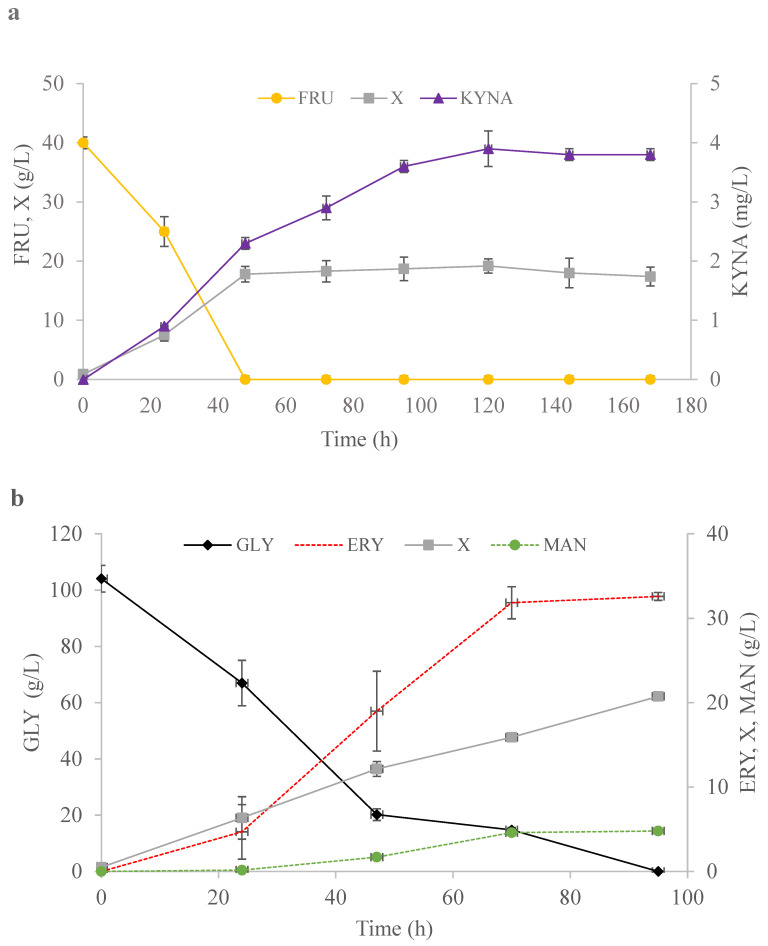
Production of: kynurenic acid by strain No. 12 (**a**), erythritol by strain No. 9 (**b**), mannitol by strain No. 5 (**c**), and citric acid by strain No. 3 (**d**) in bioreactor cultures. FRU—fructose (●), KYNA—kynurenic acid (▲), GLY—glycerol (♦), ERY—erythritol (**--**), MAN—mannitol (**--**●**--**), CA—citric acid (**--**■**--**), X—biomass (■).

**Table 1 ijms-24-07889-t001:** Characteristics of the colonies of yeast strains isolated from honey.

Strain No.	Color	Phenotype
1	light beige	wrinkled
2	light beige	wrinkled
3	light beige	wrinkled
4	light beige	wrinkled
5	light beige	wrinkled
6	light beige	wrinkled
7	light beige	wrinkled
8	light beige	smooth
9	light beige	wrinkled
10	light beige	smooth
11	light beige	wrinkled
12	light beige	wrinkled
13	light beige	wrinkled
14	light beige	wrinkled
15	light beige	smooth

**Table 2 ijms-24-07889-t002:** Comparison of identification results of selected yeast strains by API^®^/ID32 test, MALDI TOF/MS identification, and rDNA sequence-based identification.

Strain No.	Identification Profile of the Microorganism				
	API 32C Test		MALDI TOF/MS Identification		rDNA Sequence-Based Identification
	Species	The Degree of Identification, %	Species	Identification Index Value ^a^, %	Species
1	*Candida lipolytica*	99.9	*Yarrowia lipolytica*	1.90	*Yarrowia lipolytica*
2	*Candida lipolytica*	99.9	*Yarrowia lipolytica*	2.12	*Yarrowia lipolytica*
3	*Candida lipolytica*	99.9	*Yarrowia lipolytica*	2.00	*Yarrowia lipolytica*
4	*Candida lipolytica*	99.9	*Yarrowia lipolytica*	2.14	*Yarrowia lipolytica*
5	*Candida lipolytica*	99.9	*Yarrowia lipolytica*	1.99	*Yarrowia lipolytica*
6	*Candida lipolytica*	99.9	*Yarrowia lipolytica*	2.09	*Yarrowia lipolytica*
7	*Candida lipolytica*	99.9	*Yarrowia lipolytica*	2.18	*Yarrowia lipolytica*
8	*Candida magnoliae*	86.4	*Candida magnoliae*	2.13	*Candida magnoliae*
	*Candida globosa*	13.4			
9	*Candida lipolytica*	99.9	*Yarrowia lipolytica*	2.19	*Yarrowia lipolytica*
10	*Candida magnoliae*	98.7	*Candida magnoliae*	2.14	*Candida magnoliae*
11	*Candida lipolytica*	98.8	*Yarrowia lipolytica*	2.13	*Yarrowia lipolytica*
12	*Candida lipolytica*	99.9	*Yarrowia lipolytica*	1.98	*Yarrowia lipolytica*
13	*Candida lipolytica*	99.9	*Yarrowia lipolytica*	1.91	*Yarrowia lipolytica*
14	*Candida lipolytica*	99.9	*Yarrowia lipolytica*	1.97	*Yarrowia lipolytica*
15	*Candida magnoliae*	99.3	*Candida magnoliae*	1.86	*Starmerella magnoliae*

^a^ The identification index value was calculated with MALDI Biotyper 4.0.14 software (Brucker, Bruker, Poznań, Poland). Values ≥ 2.00 indicate relatedness with a high degree of confidence between the spectral sets, 1.70–1.99 indicates relatedness with a low degree of confidence between the spectral sets, and 0.00–1.69 indicates no match.

**Table 3 ijms-24-07889-t003:** Biomass productivity (Qx), maximum concentration of the main metabolite in the culture, productivity (Q_MET_), and production yield (Y_MET_) of kynurenic acid, erythritol, mannitol, and citric acid in bioreactor production cultures.

Metabolites	Strain No.	QX	Max. Concentration of Metabolite	QMET	YMET
		g/L·h	g/L	mg/L·h ^a^ g/L h	mg/g ^b^ g/g
**Kynurenic Acid Production in Bioreactor**					
Kynurenic acid	12	0.10	3.9	0.02 ^a^	0.1
Erythritol production in bioreactor					
Erythritol	9	0.22	32.6	0.34	0.33
Arabitol			1.3	0.01	0.01
Mannitol			4.8	0.05	0.05
Citric acid			1.1	0.01	0.01
Mannitol production in bioreactor					
Mannitol	5	0.21	15.1	0.16	0.15
Arabitol			1.6	0.01	0.02
Citric acid			30.2	0.25	0.30
Citric acid production in bioreactor					
Citric acid	3	0.10	75.7	0.63	0.76
Arabitol			2.05	0.17	0.02
Erythritol			4.05	0.03	0.04
Mannitol			7.15	0.06	0.07

^a^—kynurenic acid mg/L h; ^b^—kynurenic acid production mg/g.

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
