# Peer review of "Honey’s Yeast—New Source of Valuable Species for Industrial Applications"

_ijms, 2023, doi:10.3390/ijms24097889_

Round 1

Reviewer 1 Report

Great research. Some minor edits and clarification requested

The quality of presentation in English was acceptable. Those a few suggestions were made; it did not take from the overall quality of work presented.

Author Response

Dear Editor of International Journal of Molecular Sciences,

We would like to thank you and the reviewers for the your time and all the valuable comments, which helped us to submit an improved version of our manuscript “HONEY'S YEAST - NEW SOURCE OF VALUABLE SPECIES FOR INDUSTRIAL APPLICATIONS” by Patrycja Ziuzia, Zuzanna Janiec, Magdalena Wróbel-Kwiatkowska, Zbigniew Lazar, Magdalena Rakicka-PustuÅ‚ka.

Sincerely,

M. Rakicka-Pustułka

Reviewer 1

We would like to thank the Reviewer 1 for the comments. The quality of presentation in English was improved. Native speaker checked the manuscript. The responses to the Reviewer's comment are included in the manuscript file.

Reviewer 2 Report

The manuscript deals with the identification of yeast species from lime honey and analysis of their metabolites. Due to the bioreactor use for the experiments, the results may have a practical importance in industry. However, the manuscript in some parts has to be corrected. The details are listed below:

L11-12: start the statement that honey is a source of compounds with biological activity but also is a source of microorganisms

L27-36: Add a first paragraph related to the importance and properties of honey. Include composition of vitamins, phenolic acids, flavonoids, proteins, carbohydrates, amino acids, royal jelly aliphatic acids. For this purpose the Authors may refer to the following references: https://doi.org/10.2478/jas-2018-0012, https://doi.org/10.3390/ijerph20032458.

L116: fifteen yeast strains

L119: in point 2 of the Fig. 1 include also DNA sequencing as a method of yeasts identification in this study

L169: in this paragraph the discussion is missing. Indicate also % differences in growth between isolates

L191-193: transfer to Materials and methods

L213: In Fig. 3a kynurenic acid

L245-247: rephrase

L290: it is 32.6 g/L in Table 3

L328: indicate these values or % differences related to the reference

L332: was it one honey sample?

L452, 459: add injection volume and time of analysis

Moderate editing of English is needed.

Author Response

Dear Editor of International Journal of Molecular Sciences,

We would like to thank you and the reviewers for the your time and all the valuable comments, which helped us to submit an improved version of our manuscript “HONEY'S YEAST - NEW SOURCE OF VALUABLE SPECIES FOR INDUSTRIAL APPLICATIONS” by Patrycja Ziuzia, Zuzanna Janiec, Magdalena Wróbel-Kwiatkowska, Zbigniew Lazar, Magdalena Rakicka-PustuÅ‚ka.

Sincerely,

M. Rakicka-Pustułka

Reviewer 2

We would like to thank the Reviewer 2 for the comments. The quality of presentation in English was improved. Native speaker checked the manuscript. The responses to the Reviewer's comment are listed below.

The manuscript deals with the identification of yeast species from lime honey and analysis of their metabolites. Due to the bioreactor use for the experiments, the results may have a practical importance in industry. However, the manuscript in some parts has to be corrected. The details are listed below:

We would like to thank the Reviewer 2 for the comments.

L11-12: start the statement that honey is a source of compounds with biological activity but also is a source of microorganisms

The sentence was changed: “Honey is a rich source of compounds with biological activity moreover; it is also a valuable source of various microorganisms.”

L27-36: Add a first paragraph related to the importance and properties of honey. Include composition of vitamins, phenolic acids, flavonoids, proteins, carbohydrates, amino acids, royal jelly aliphatic acids. For this purpose the Authors may refer to the following references: https://doi.org/10.2478/jas-2018-0012, https://doi.org/10.3390/ijerph20032458.

The introduction of the manuscript was improved and the following information was included:

“Honey is largely composed of glucose and fructose (about 75%), however, the quantitative ratio of these two sugars is often an indicator of honey origin. In most types of honey, the amount of fructose slightly exceeds the content of glucose. The exception to this rule are grape’s honey (Brassica napus) as well as honey obtained from daffodils (Taraxacum officinale) (Da Silva et al. 2016). The second most abundant component of honey is water, which is a factor determining the occurrence of microorganisms. The water activity in honey oscillates at the level of 0.5-0.65 (Da Silva i wsp. 2016). Moreover, the protein content and qualitative composition of amino acids varies depending on the origin of honey (Zamora and Chirife, 2006). The most common amino acid is proline (50-85% of the total amount of amino acids), followed by glutamic acid, alanine, phenylalanine, tyrosine, leucine and isoleucine (Girolamo et al. 2012, Da Silva et al. 2016). Among the organic acids found in honey the highest concentration (approx. 0.57%), was measured for gluconic acid, which significantly affects the pH of the honey, which ranges from 3.2 to 4.5 (Da Silva et al. 2016). Potassium (K) is the main mineral component found in honey, but significant amount of sodium (Na), iron (Fe), copper (Cu), silicon (Si), manganese (Mn), magnesium (Mg) and calcium (Ca) have also been reported in the literature (Da Silva et al. 2016). Honey also contains vitamin C and a complex of B vitamins (B2, B3, B5, B6, B8, B9), which are protected from oxidative degradation due to the low pH of this product (Da Silva et al. 2016). Honey has antibacterial and antifungal properties which are mostly caused by the low water activity, presence of gluconic acid as well as the presence of fatty acids originating from royal jelly, the larval food of honeybees (Isidorow et al., 2018). The aliphatic C8-C12 acids, identified in honey samples, demonstrated antibacterial action against different microbes, and those are the main components exhibiting the antibacterial activity of honey (Isidorow et al., 2018). Another compounds with this activities in honey are phenolic acids, present in the highest concentration in phacelia (356.72 µg/g) and multifloral honeys (318.9 µg/g) (Kunat-BudzyÅ„ska et al., 2023).”

L116: fifteen yeast strains

The word was changed.

L119: in point 2 of the Fig. 1 include also DNA sequencing as a method of yeasts identification in this study

The Fig. 1 was improved as suggested by the Reviewer 2.

L169: in this paragraph the discussion is missing. Indicate also % differences in growth between isolates

We are grateful to the Reviewer 2 for this comment. The section was improved as follows:

“Strain No. 8 achieved the highest growth on glucose, while for strains No. 10 and 15 the growth was 7% lower compared to the strain No. 8 on the same substrate. The remaining strains obtained the lower growth as much as 28-40% of the maximum growth reached for strain No. 8 growing on glucose (Figure. 2a). The analyzed Y. lipolytica strains isolated in this study showed weaker growth on glucose than Y. lipolytica strain A101 described by Dobrowolski et al., 2019 (48). Considering the second most abundant sugar - fructose, the strains with the highest growth were No. 8, 10 and 15 (Figure. 2b). In contrary, strains No. 4, 7 and 11 showed the lower growth by 41-48%, compared to the strain No. 8 growing on fructose (Figure. 2b). The observed differences in growth between glucose and fructose may be connected to high oxygen demand for Y. lipolytica, while oxygen availability during microcultures is limited. The Candida/Starmerella magnoliae isolates No. 8, 10 and 15 grew well also on inulin and sucrose (Figure 2c, d). The other isolates not growing on inulin and sucrose (Figure 2c, d), what confirms the general knowledge about Y. lipolytica strains, that they are not able to use sucrose and inulin as a sole carbon source (49). The most significant differences were observed for the analyzed set of strains growing on technical glycerol (Figure. 2e). Strain No. 1 did not grow at all in this medium. The isolate No. 10 showed the best growth and reached the highest biomass (Figure. 2e). For strains No. 2-5, 7-9, 11-12 and 15, the growth on technical glycerol was lower by 15-20%, compared to the strain No. 10. Strains No. 6 and 13 showed longer lag phase compared to other isolates and the lowest growth reached at the end of the culture (22-37% lower than for strain No. 10). The growth of Y. lipolytica strains No. 2-5, 7-9 and 11-12 on technical glycerol was similar to the growth of other Y. lipolytica strains described by MiroÅ„czuk et al., 2017 (50).”

L191-193: transfer to Materials and methods

The sentences were transferred to the Materials and methods section.

L213: In Fig. 3a kynurenic acid

The word was changed.

L245-247: rephrase

The sentences were changed as follows:

“Citric acid is produced by Y. lipolytica under limited nitrogen availability (31). Surprisingly, in CAMMed medium used in this study, the analyzed strains did not produced high amounts of citric acid, however, they produced lot of polyols instead (Figure. 4C). The CAMMed medium composition consist of high quantity of yeast extract (3 g/L), which might favor the biosynthesis and secretion of polyols by the tested yeast. In the following studies, it would be worth to assess the effect of yeast extract concentration on citric acid/polyols synthesis by the chosen strains as well as optimize the medium composition and culture condition for the particular strain.”

L290: it is 32.6 g/L in Table 3

The correct amount is 32.6 g/L.

L328: indicate these values or % differences related to the reference

The section was improved as follows:

“The lowest reported citric acid yield for Y. lipolytica growing on glucose was noted for the NBRC 1658 strain (0.38 g/g) during batch culture and for H222 strain (0.39 g/g) growing in fed-batch culture (61). In contrary, the highest citric acid yield was reached for different Y. lipolytica strains growing on glucose: A-101 (repeated batch mode), W29 (batch culture) and A-101-1.14 (batch culture) and reached 0.84, 0.85 and 0.93 g/g, respectively (61). The citric acid yield obtained in this study for Y. lipolytica strain No. 3 during batch culture on technical glycerol (0.76 g/g), was higher than the highest yield (0.62 g/g) described in the literature for Y. lipolytica 1.31 growing on raw glycerol in batch culture (61).”

L332: was it one honey sample?

The sentence was changed: “Lime honey samples were obtained aseptically … “

L452, 459: add injection volume and time of analysis

The information has been added. “The injection volume was 10 µl and the run time lasted 15 min.”

Moderate editing of English is needed.

Native speaker checked the manuscript.

Round 2

Reviewer 2 Report

Authors have improved the manuscript. I have no more comments.